

# The direct and indirect effects of a global pandemic on US fishers and seafood workers

Easton R. White[1,2], Jill Levine[3,4], Amanda Moeser[5] and Julie Sorensen[6]

[1] Department of Biological Sciences, University of New Hampshire, Durham, NH, United States
[2] Gund Institute for Environment, University of Vermont, Burlington, VT, United States
[3] Department of Forestry, University of Vermont, Burlington, VT, United States
[4] Department of Biology, University of Vermont, Burlington, VT, United States
[5] Environmental Studies Department, Antioch University New England, Keene, NH, United States
[6] Northeast Center for Occupational Health and Safety: Agriculture, Forestry and Fishing, Bassett Healthcare Network, Cooperstown, New York, NY, United States

Corresponding author
Easton R. White,
eastonrwhite@gmail.com

## ABSTRACT

The United States' fishing and seafood industries experienced major shifts in consumer demand and social-distancing restrictions starting in March 2020, when the early stages of the COVID-19 pandemic were unfolding. However, the specific effects on fishers and seafood processors are less well known. Fishermen and seafood workers are potentially at risk during a pandemic given existing tight working quarters, seasonal work, and long hours. To address these concerns, and given a lack of data on the sector, we reviewed news articles, scientific articles, and white papers to assess the various effects of COVID-19 on US seafood workers. Here, we show that most COVID-19 cases among seafood workers occurred during summer 2020 and during the beginning of 2021. These cases were documented across coastal areas, with Alaska experiencing the largest number of cases and outbreaks. Seafood workers were about twice as likely to contract COVID-19 as workers in other parts of the overall US food system. We also documented a number of indirect effects of the pandemic. New social-distancing restrictions and policies limited crew size, resulting in longer hours and more physical taxation. Because of changes in demand and the closure of some processing plants because of COVID-19 outbreaks, economic consequences of the pandemic were a primary concern for fishers and seafood workers, and safety measures allowed for seafood price variation and losses throughout the pandemic. We also highlight a number of inequities in COVID-19 responses within the seafood sector, both along racial and gender lines. All of these conditions point to the diverse direct and indirect effects of the COVID-19 pandemic on fishers and seafood workers. We hope this work sets the foundation for future work on the seafood sector in relation to the COVID-19 pandemic, improving the overall workplace, and collecting systematic social and economic data on workers.

## INTRODUCTION

The seafood industry is well-known for its high rates of occupational injury and fatality. While work-related injuries have been a focus in the literature on seafood workers, several articles have indicated that work-related illness is still a cause for concern for workers across seafood value chains (*Sorensen, Echard & Weil, 2020*; *White et al., 2021*). According to fatality rates, injuries, and comorbidities among fishing industry workers were reported to be higher than in other industries even before the spread of coronavirus in the U.S. The average fatality rate among all US full time workers was four deaths per 100,000 from 2000 to 2017 compared to the US commercial fisher fatality rates of 114 deaths per 100,000 full-time workers in 2000–2017 (*BLS, 2019*). U.S. Coast Guard and OSHA reports also indicate that the Alaskan seafood processing industry is "high-risk" for traumatic injuries (*Syron et al., 2018*). It was estimated that the fatality rate for seafood workers in Alaska was 121 for every 100,000. This is 34% higher than the average rate of all U.S. workers (*Lucas et al., 2014*). Coast Guard injury reports have underscored the need to prevent musculoskeletal injuries to the upper body and extremities, as more serious incidents (back injuries, intracranial injuries, finger crushing, amputations) can lead to disability. Post-traumatic stress disorder (PTSD) and its symptoms have also been reported by fishers and seafood workers, with some fisheries workers in the Northeast reporting significantly higher rates of PTSD than other U.S. workers. These symptoms include intrusive memories, anger, and irritability. Additionally, fishers endure a higher prevalence of sleep apnea, hearing loss, and musculoskeletal disorders, all having the potential to lead to addiction to pain medications and overdose deaths (*Sorensen, Echard & Weil, 2020*).

Several studies of the impact of the COVID-19 pandemic on the seafood sector, indicated that COVID had both direct impacts to health as well as indirect effects connected to reduced labor access and economic losses (*Bennett et al., 2020*; *Knight et al., 2020*; *Love et al., 2021a*; *Ross, Jacobs & Oliver, 2021*; *White et al., 2021*). Even before the appearance of COVID cases in the US, overseas demand for seafood products dropped due to pandemic lockdowns in China (*Gephart et al., 2020*; *White et al., 2021*). With the worldwide spread of the pandemic, global demand for seafood was significantly reduced (*FAO, 2020*; *Love et al., 2021a*; *White et al., 2021*), greatly impacting U.S. fisheries given the country's position as a major trader of seafood (*Gephart & Pace, 2015*). In addition, pre-COVID, restaurants accounted for 65% of consumer expenditures for seafood (*Love et al., 2020*). Thus, social-distancing restrictions and onsite dining restrictions greatly affected demand for seafood products. A 2021 NOAA report noted that commercial fisheries landings declined 22% in 2020 as compared to the previous five year period (*NMFS, 2021*). In a survey of 260 commercial fishers, authors found a considerable increase in fishers work hours as they attempted to switch to direct consumer sales (*Smith et al., 2020*). Declines in profit also led to smaller crews, which meant longer work hours for those still fishing. Forty-five percent of fishers who tried direct consumer marketing reported more than a 20% decline in income (*Sorensen, Echard & Weil, 2020*).

With a loss of income, fishing boat captains likely had less money to invest in safety, which can increase the potential for hazardous exposures on fishing vessels. U.S. Coast Guard inspectors found a rise in safety violations on vessels from 10% to 30% in their district since the COVID-19 pandemic, as well as an increase in the number of boats sent back to port for unsafe conditions (*Sorensen, Echard & Weil, 2020*). Fishing and processing vessel crew quarters are tight spaces even on large ships. Crew members typically share sleeping quarters (bunk beds) and eating areas are communal by necessity. In these circumstances, social distancing is largely impossible and use of personal protective equipment, such as masks, is also problematic, given environmental conditions on fishing boats (wind, water, etc.). Vessels are often constantly wet and therefore make personal protective equipment (PPE) less sanitary and more hazardous to human health due to the accumulation of bacteria from fish (*Syron et al., 2018*). The risk of COVID-19 outbreaks are significant in high-density workplaces, such as seafood processing plants. Alaska's seafood processing facilities attract almost 18,000 out-of-state workers every year. According to the state's reportable disease database (*Porter et al., 2021*), 667 cases of COVID-19 were identified among seafood processing workers. One-hundred and two of those cases were independent, occurring from either quarantine groups in or outside of the facilities, however, the remaining 539 cases were part of outbreaks that spread beyond quarantine groups, both within and outside of the facility. Six of the cases identified in the state's database did not have enough information to determine transmission details. These outbreaks led Alaskan state personnel and CDC field employees to increase COVID-19 restriction measures by mandating reductions in group gatherings to less than 10 individuals. These authorities also noted that "safe" transfer of crews midseason was an inadequate measure for preventing the spread of the virus, as there was an outbreak during the transfer of a crew from one processing facility to another (*Porter et al., 2021*). Entry testing and quarantining were found to help prevent the transmission of COVID-19 from processing plants and vessels.

While several peer-reviewed manuscripts have explored the impact of the global COVID-19 pandemic on the seafood sector, our goal was to document the spread of COVID-19 within the US seafood sector and also highlight indirect effects of the pandemic on U.S. seafood workers. Specifically, we examine news articles and data on COVID-19 cases in the seafood industry to understand when and where COVID-19 cases occurred, the indirect and direct effects of the pandemic throughout the seafood value chain, and how the seafood sector fared compared to similar industries. We hypothesize that most cases and outbreaks would happen early during the pandemic, seafood workers would be disproportionally affected by the pandemic compared to other non-seafood workers, and seafood workers would experience several indirect effects as a result of the pandemic.

## MATERIALS AND METHODS

### News articles

We examined news articles to see how the media was reporting on stories related to COVID-19 and the seafood industry. We describe a few specific case studies below, but we

also examined reporting over time using the Global Database of Events, Language and Tone (GDELT) of news articles (*GDELT, 2021*). We restricted our search to articles published in the US from January 2020 to September 2021. Specifically, we examined the frequency of articles that included the terms (COVID) AND (worker OR employee) AND (seafood OR fisheries). This search criteria added up to 20,229 individual news articles, which includes articles that might be focused on the same event. This total also includes some articles published in the US that were focused on events outside the US as well as articles that might not focus on our specific topic. We did not filter articles beyond this point, mostly due to the large number of articles and limitations of the GDELT database, *e.g.*, not being able to filter out articles that refer to the same event.

## COVID-19 cases in the seafood industry

Throughout the COVID-19 pandemic, cases and outbreaks of COVID-19 among seafood workers were reported sporadically by news outlets (*Douglas, 2021*; *Korban & Cherry, 2020*; *White et al., 2021*). However, there were very few localities (*e.g.*, local health department or hospital) that reported COVID-19 cases broken down by industry or business. We used a compilation of news articles to build a database of cases within the seafood industry specifically. This was primarily facilitated using data collected by The Food and Environment Reporting Network (FERN, *Douglas, 2021*). In cases where the FERN database had missing values (*e.g.*, missing values for deaths), we added information based on additional searches and previous studies. The data includes cases and deaths among workers across the US food system, including farms, food processors, and meatpacking. Included within the FERN data are cases among seafood workers in processing and distribution facilities as well as on fishing vessels. The data also includes information on locations, industry, and company. We defined each record in the database as an "outbreak" regardless of the exact number of cases. We examined the number of cases and deaths among fishers and seafood processors throughout the COVID-19 pandemic. Specifically, we examined how cases varied over time, by location, and across the seafood value chain. We used generalized linear models with a Poisson error distribution to assess whether cases and outbreaks have decreased over time. We verified model assumptions with standard residual plots.

We also compared (using a Chi-square test) how caseloads in the seafood industry compared to other related industries, such as farming and meatpacking. Specifically, the U.S. Bureau of Labor Statistics noted that in May 2020, 1,596,500 and were employed in the overall food manufacturing (NAICS 311000) and 29,290 workers seafood product preparation (NAICS 311700) sectors, respectively (*U.S. Bureau of Labor Statistics, 2021*). Food manufacturing includes a wide array of sectors, including meat processing, factory workers, and seafood packaging. The employment numbers also include everything from top executives to front line workers. Seafood Product Preparation and Packaging includes fish butchers and processors. Similarly, USDA data from 2018 show that of 1.7 million people employed in "U.S. food and beverage manufacturing", only 1.9% (or 32,300 individuals) were employed in seafood (*USDA, 2018*). Similarly, a 2018 NOAA report indicated that there were approximately 30,708 U.S. workers in seafood preparation

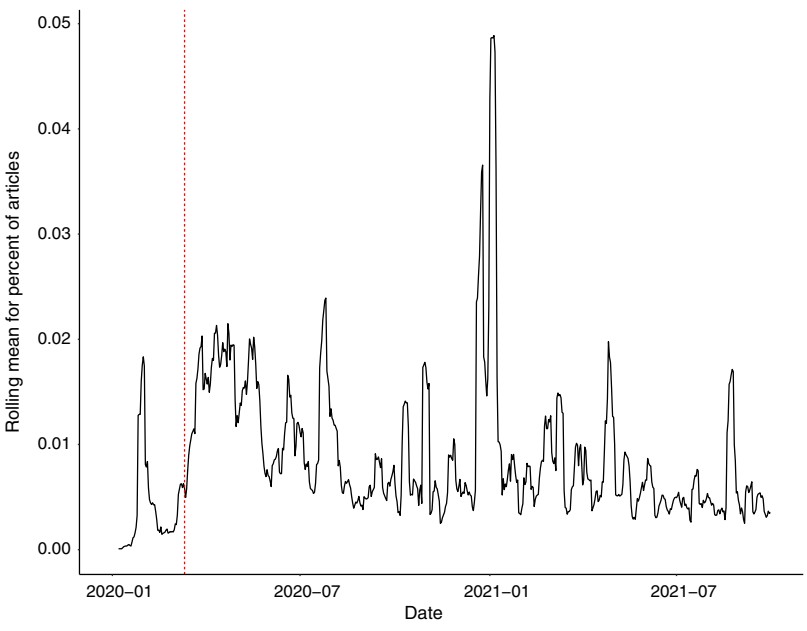

**Figure 1 Rolling mean (weekly average) for the percent of total news articles (*GDELT, 2021*) mentioning (COVID) AND (worker OR employee) AND (seafood OR fisheries). The vertical red line denotes the date (11 March 2020) when the World Health Organization declared a global pandemic.**

and packaging (*National Marine Fisheries Service, 2018*). Thus, all three sources show a similar two-fold difference between the number of seafood workers *versus* those in food manufacturing more generally.

## RESULTS

### News articles

Between March 2020 and September 2021, we identified a total of 20,229 US news articles that specifically mentioned (COVID) AND (worker OR employee) AND (seafood OR fisheries) (Fig. 1). We found that most of the news coverage around COVID-19 and seafood workers occurred during the beginning of the pandemic and in early 2021 (Fig. 1). The coverage mostly included information on migrant seafood workers, vaccination campaigns, and outbreaks. The coverage also included articles focused on new policies implemented to combat the pandemic, such as financial assistance programs, limits on personnel on vessels, or new PPE requirements. In May 2020, sole proprietors, fishers, and some contract workers became eligible to apply for a new federal Pandemic Unemployment Assistance (PUA) program designed to extend unemployment benefits to self-employed workers (*Maine Coast Fishermen's Association (MCFA), 2021*). In addition, $300 million was awarded to states, tribes, and territories with inland and coastal fisheries to provide relief funds, including direct payments, to fisheries and aquaculture participants through the Coronavirus Aid, Relief, and Economic Security Act (CARES Act) (*NOAA, 2021*). At the same time, in June 2020, the CDC, OSHA, and FDA reported that seafood processing workers were at increased risk of exposure to COVID-19 and published

an interim guidance document with specific recommendations on protecting seafood workers from COVID-19 (*OHS, 2020*). However, relief programs designed to reimburse seafood processing companies for investments in workplace health and safety were not announced until the following year and funds have yet to be distributed to seafood processors. In September 2021, USDA announced $650 million in funding to support the Pandemic Response and Safety Grant program, which includes eligible seafood processing facilities and vessels, and $50 million in additional funds available to seafood processors to enhance workplace safety measures, retrofit facilities, provide PPE, and cover medical costs associated with COVID-19, such as vaccination, testing, and paid sick leave (*USDA, 2021*). In September 2021, some seafood companies also started requiring workers to be vaccinated for COVID-19.

## COVID-19 cases in the seafood industry

The first reported seafood worker COVID-19 outbreak in the US occurred in April 2020 at a processing facility in the Pacific Northwest. By May 2020, additional seafood worker outbreaks had been reported across the country—from the Northeast to the Gulf Coast—primarily in seafood processing plants, but also at crawfish farms in Louisiana (*Korban & Cherry, 2020*). Cases of COVID-19 among seafood workers were reported or documented in 13 states, including Maine, New Hampshire, Massachusetts, Rhode Island, Pennsylvania, Virginia, Maryland, North Carolina, Louisiana, California, Oregon, Washington, and Alaska. Most cases and outbreaks across all food systems were observed in April and May of 2020 and have decreased since (Figs. 2 and 3). In the seafood industry, outbreaks and cases were more sporadic with peaks occurring between June–August 2020 and January–February 2021 (Fig. 2). With the seafood sector, outbreaks were documented in seafood plants and processing, seafood distributors, seafood wholesalers, aquaculture farms, and on vessels. As expected, most seafood-related COVID-19 cases and outbreaks occurred in coastal areas, with Alaska at the top of the list (Fig. 3). In addition, our numbers on COVID-19 cases were similar to a paper focused on Alaska seafood workers (*Porter et al., 2021*). Specifically, between March 1st and October 13th of 2020, we noted 496 seafood-related COVID-19 cases in Alaska compared to an independent estimate of 539 (*Porter et al., 2021*).

Despite being a small part of the overall food system, the seafood industry accounted for 3.84% and 5.23% of all cases and outbreaks, respectively, although it is difficult to obtain exact employment numbers to compare industries. As an estimate, we used the total number of those employed in Food Manufacturing (NAICS 311000) in the US (May 2020 = 1,596,500) and the total number of people employed in Seafood Product Preparation and Packaging (NAICS 311700), which was 29,920 (*U.S. Bureau of Labor Statistics, 2021*). Thus, there was a two-order magnitude difference in employment for seafood *versus* the food system overall. As a rough estimate, we estimated there were 65 COVID-19 cases per 1,000 workers in seafood *versus* 31 cases per 1,000 for the overall food system. These numbers are difficult to compare to the general population as the cases identified here in food systems include only publicly documented cases or outbreaks for workplaces, as opposed to cases in general.
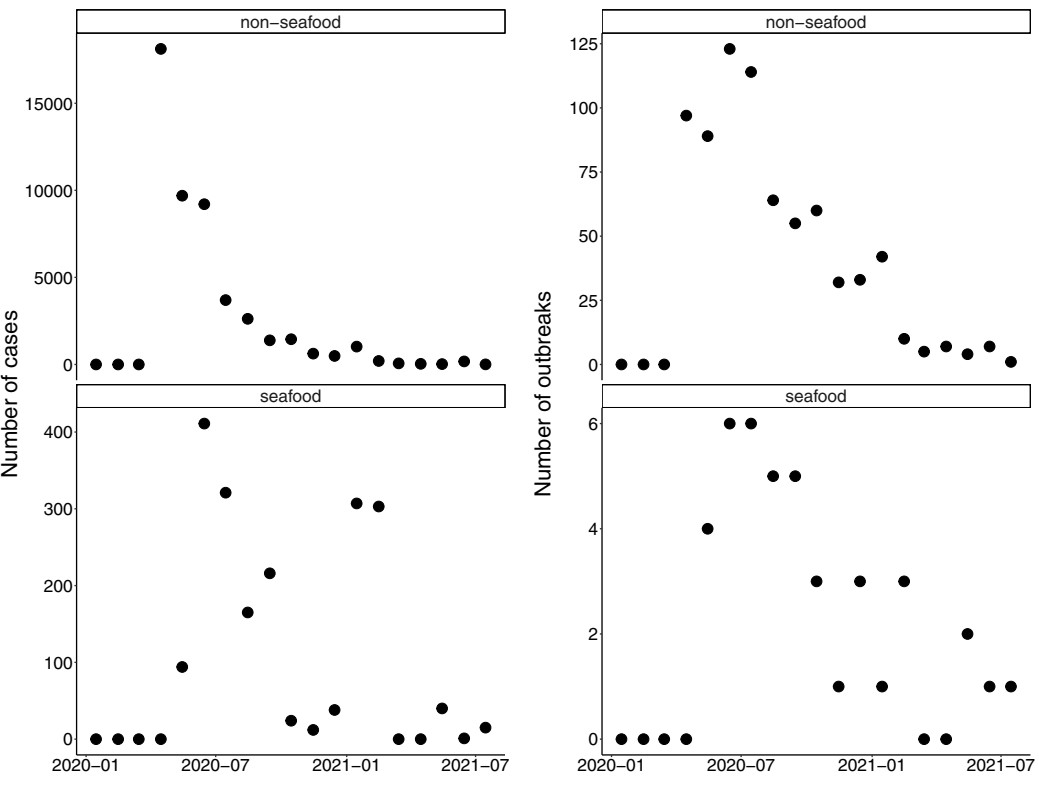

**Figure 2 Total monthly COVID-19 cases and outbreaks (i.e., number of events reported) for non-seafood (e.g., meatpacking, food processing, and farms) and seafood workers. There was a significant decrease in cases over time for both non-seafood ($\beta = -0.44$, $p < 0.001$) and seafood workers ($\beta = -0.10$, $p < 0.001$). There was also a significant decrease in outbreaks over time for both non-seafood ($\beta = -0.20$, $p < 0.001$) and seafood workers ($\beta = -0.10$, $p = 0.005$).**

## DISCUSSION

We found that both news coverage and documented COVID-19 cases showed similar patterns for seafood workers (Figs. 1 and 2). We found peaks in seafood-related COVID-19 cases during summer 2020 and in early 2021 (Fig. 2). There was a significant decrease in cases and outbreaks over the course of the study (Fig. 2), but the relationship was stronger for non-seafood workers. These cases occurred primarily in coastal areas, with the most cases, and cases *per capita*, occurring in Alaska (Fig. 3). Seafood-related COVID-19 cases occurred throughout the seafood value chain, including fishing vessels, seafood processors, and seafood distributors. We also found that seafood workers were more likely (Table 1) to contract COVID-19 compared to non-seafood workers. News coverage showed similar patterns with a focus on outbreaks in processing facilities and vessels, but also documented indirect effects of the pandemic. These indirect effects included financial losses reported by fishers, aquaculturists, processors, and industry groups, in addition to changes in market demand, social distancing and travel restrictions, supply chain issues, and worker shortages.

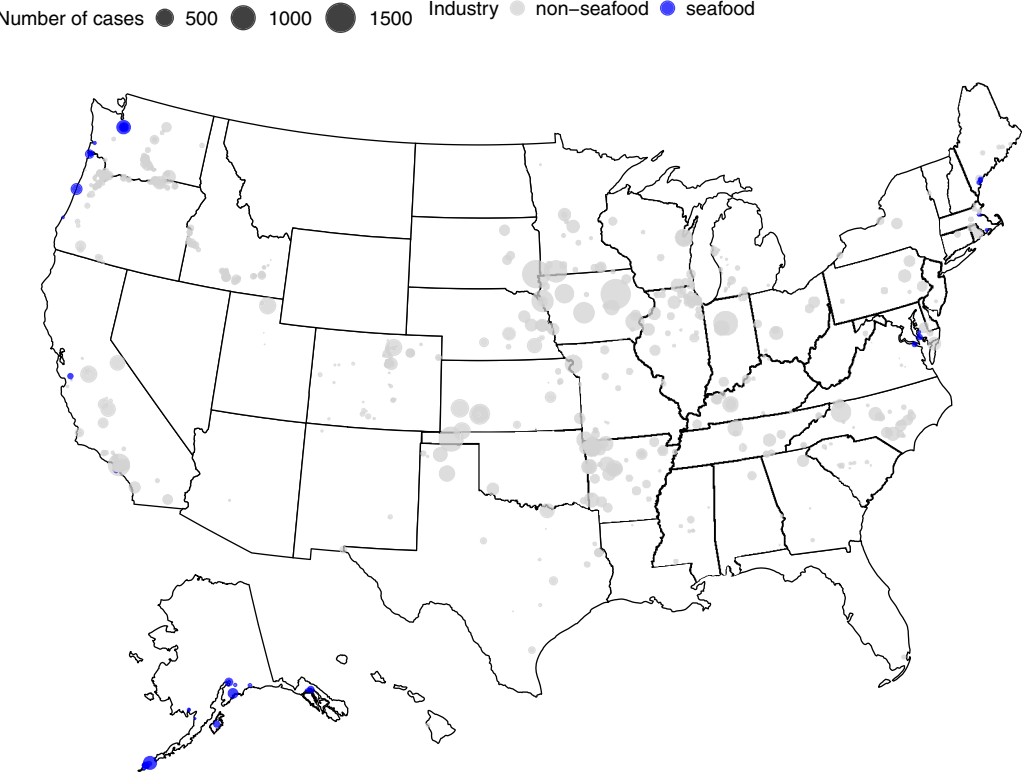

**Figure 3 Spatial distribution of COVID-19 cases across the food system industry. The data is compiled from news articles from April 2020 to July 2021 (*Douglas, 2021*). The light grey points denote non-seafood cases and the blue points highlight seafood-related cases.**

**Table 1 Number of cases and outbreaks by food industry sector.**

| Industry | Cases | Outbreaks | Cases per 1,000 workers | Outbreaks per 1,000 workers |
|---|---|---|---|---|
| Non-seafood | 48,799 | 743 | 31.15 | 0.47 |
| Seafood | 1,947 | 41 | 65.07 | 1.37 |

Notes:

The total number of workers for each sector (for May 2020) come from the U.S. Bureau of Labor Statistics (*U.S. Bureau of Labor Statistics, 2021*) with 1,596,500 and 29,290 employed in the overall food manufacturing (NAICS 311000) and seafood product preparation (NAICS 311700), respectively. There was a significant difference in cases *per capita* for non-seafood *versus* seafood workers ($\chi^2$ = 12.04, $p$ = 0.00052).

In addition to COVID-19 outbreaks, deaths, and safety concerns, seafood workers faced several indirect effects from the COVID-19 pandemic such as loss of income, changes in consumer spending, and supply chain issues (Fig. 1; *Love et al., 2021b*; *Ross, Jacobs & Oliver, 2021*; *Stoll et al., 2021*; *White et al., 2021*). There was limited information on how the pandemic has affected seafood workers' mental and physical health, and whether more accidents and fatalities arose because of the pandemic. However, more has been studied on the economic side, as the seafood industry has spent over $50 million to reduce the spread of COVID-19 (*Ross, Jacobs & Oliver, 2021*). At the beginning of the coronavirus pandemic, 70% of farms and businesses in the North Central Aquaculture region farms and businesses reported loss of sales (*Senten & Smith, 2020*). For the fourth quarter of the

2020 year, focusing on mollusks and fish food, researchers found that 83% of respondents' farms had been impacted and 61% believed their farms would continue to be impacted in 2021 (*Senten & Smith, 2020*). News articles have reported lockdowns and curfews reducing the sizes of catches and forcing reducing fishery worker income to make less pay per day (*Quallen, 2021*). A Oceanic Strategies survey conducted with 400 fishers and reported by the Alaska Journal of Commerce found 98% believed their business was "bashed" by the pandemic (*Welch, 2021*). A total of 82% said fishing was their primary source of income, and 91% claimed they lost 15–100% of their revenues since January 2020. It was also reported that restrictions and measures to handle the pandemic had an immediate effect on the fishing market. Another survey of 200 fishers, found that one-third claimed they were unsure they would be fishing in 3 years (*Zimmer, 2021*).

Alaska and the town of New Bedford, Massachusetts present an interesting comparison to highlight the variety of responses to the pandemic (Panel 1). Previous work, and this paper, have shown the high number of seafood-related COVID-19 cases in Alaska (*Porter et al., 2021*). Alaska represents 60% of the United States commercial fisheries with 40% of the nation's surface water (*Liddel & Yencho, 2021*; *Resource Development Council, 2022*). In 2019, Alaskan commercial fisheries produced 5.6 billion pounds of seafood worth $1.7 billion, with the two largest commercial fisheries being Dutch Harbor and Aleutian Islands (*Liddel & Yencho, 2021*). As of Fall 2021, there have been 17 outbreaks, resulting in 848 cases in Alaska seafood alone. We hypothesize this is largely the result of several interconnected factors. First, seafood is a major industry in Alaska accounting for at least 1 million metric tons of food each year and over $1.8 billion of labor income with 60,000 workers and greater than $5 billion income for the U.S. economy (*McDowell Group, 2017*). Seafood processors reported their biggest challenge throughout the pandemic has been finding employers for their plants and that they spent over $50 million on COVID-19 prevention measures in 2020, including food costs, coronavirus testing, and flying employees to Alaska and quarantining them in hotels before the season starts (*White, 2020*). Second, a lot of the labor for Alaskan fisheries is seasonal and involves migrant workers (*Food and Agriculture Organization of the United Nations, 2020*; *Sánchez Pulla et al., 2021*). In 2019, 77% of seafood's processing industry was made up of migrant workers, with 90% of workers being non-residents in the Aleutian Islands. In 2020, 91.2% of the Aleutian Island workers were nonresidents (*Alaska Department of Labor and Workforce Development, Research and Analysis, 2020*; *Krieger & Whitney, 2021*). Although a large amount of seafood processors in Alaska are non-resident or migrant workers, the H-2B program of the U.S. Department of Labor did not provide these workers with COVID-19 protection rules or measures (*Sánchez Pulla et al., 2021*). Cumulative rates of COVID-19 in Alaska for nonresidents is 24% of resident cases, with nonresidents accumulating 5,145 cases and residents accumulating 124,568 cases. Between September 16 and October 27, there were 752 nonresident reported cases and 38,662 resident reported cases (*Alaska Department of Health & Social Services, 2021*). This concern is partially due to seafood processors failing to engage in proper Department of Labor COVID-19 safety measures. For example, in March of 2021, it was reported that Copper River Seafoods violated COVID-19 safety precautions by failing to effectively

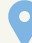 **New Bedford, Massachusetts**

The town of New Bedford, Massachusetts has been to the United States' wealthiest fishing port for almost two decades now. This port, specifically on pier 3, was the first location to set up a COVID-19 testing center. It opened in March 2020, while the rest of the country needed proof of symptoms or high-risk travel to report a case. Each crew, using the bubble system, would test in between fishing trips and was asked to quarantine for 2-3 days until test results would come back. With this system, only four vessels tested positive, and each was contained before an outbreak started. As of April 2021, New Bedford waterfront opened a Johnson and Johnson vaccine clinic. As of April 13th, over 1000 people were vaccinated and 1,200 individuals had appointments.

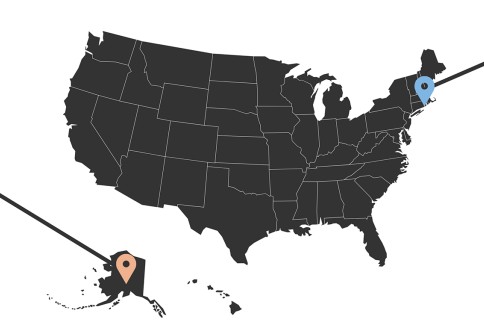

**Alaska**

Alaska experienced the largest number of seafood-related COVID-19 outbreaks throughout the pandemic (McKenney, 2021b). For example, on Aleutian island in Akutan, Trident Seafoods had to shut down due to 700 COVID-19 cases among workers in early 2021. Because the processing plant is on an island, no planes were able to fly for rescue, and there was limited supplies due to bad weather. This inhibited fishermen from unloading their boats, specifically of cod. The other large processing plant in this region is the Bering Sea, which has had outbreaks in Unalaska, Alaska. It is evident that Alaska had trouble managing the spread of the virus, as in early fall 2020, 677 new cases were reported via 13 separate seafood processing facilities or processing vessel outbreaks (Porter et al., 2021).

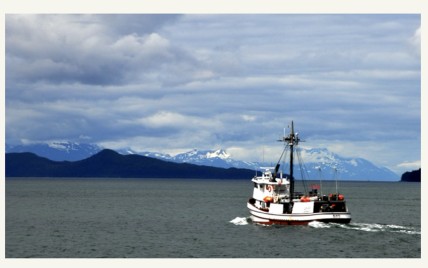

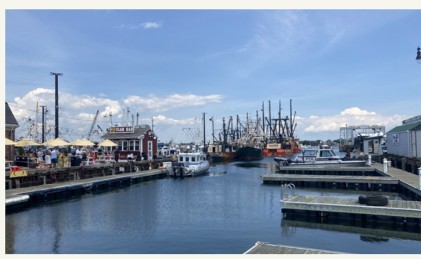

**Figure 4 Case studies of responding to the COVID-19 pandemic.**

screen employees for COVID-19, implement social distancing, prevent sick employees from work, and failure to input physical barriers for employees (*Maguire, 2021*; *Field et al., 2021*). Copper River Seafoods was ultimately fined $134,937 plus $241,610 in fees for COVID-19 workplace safety violations after 70% of their employees contracted COVID-19. Comparison to other states, Alaska either lagged, or did not implement, many social distancing restrictions (*White & Hébert-Dufresne, 2020*; *Althouse et al., 2020*) and has a low vaccination rate as of Fall 2021 (*Dong, Du & Gardner, 2020*). And lastly, logistics in Alaska are difficult in terms of acquiring supplies or providing medical care.

We can contrast this with the town of New Bedford, Massachusetts which contains a massive shipping port where about 390 million pounds of fish pass through annually. It has been the United States' wealthiest fishing port for almost two decades now. The town and the state were more preemptive in responding to COVID-19 with a particular emphasis on the seafood sector. The town was one of the first to open both testing centers and vaccination sites for seafood workers specifically. Consequently, the town has seen only a limited number of cases and outbreaks in comparison to those in the Greater New Bedford area (*New Bedford Health Department, 2021*) (Fig. 4). While New Bedford, Massachusetts was a leader among port towns, the town was found to be behind the rest of Massachusetts state-wide vaccination rate average, with a wider gap for minority groups. As of November 2, 2021, the state-wide vaccination rate average was 68.1%, while New Bedford's town average was 50.2% (*Mass.gov, 2021*). Within New Bedford, 52.1% of

the white population is vaccinated, 58.5% of the black population, 31.8% of the multiracial population, 55.2% of the Asian population, 34.1% of the American Indian/Alaska Native and 51.8% of the Hispanic population. The vaccine hesitancy is reportedly due to misinformation and a lack of education from trustworthy individuals within identical communities (*Spillane, 2021*). There is also allegedly a lack of time in daily schedules and access to health care for certain individuals. Certain families have one individual working long day shifts, with the other partner working the opposite long shifts.

Available data and associated news coverage suggest inequity in terms of COVID-19 impacts and response due to occupational segregation based on gender, race, and immigration status within the seafood industry. A review of media coverage, as well as financial relief programs, policy, and research in response to the COVID-19 pandemic appear to prioritize participants in the commercial fishing and aquaculture sectors. However, available evidence suggests workers in the processing sector were more likely to be directly impacted by COVID-19 as outbreak events were concentrated in processing plants and at-sea processing vessels. It is crucial to note that twice as many people are employed in seafood processing and distribution than commercial fishing and most of these workers are women, minorities, and immigrants (*New American Economy Research Fund, 2017*; *National Guestwork Alliance, 2016*). Black and Hispanic workers are more than twice as likely as white workers to be employed in animal processing (*Hawkins, 2020*) and foreign-born workers are known to play a particularly important role in the U.S. seafood processing sector (*New American Economy Research Fund, 2017*). In 2019, 61.7% of workers in the seafood processing sector were foreign-born (*New American Economy Research Fund, 2017*). Furthermore, nearly 13,000 workers were granted temporary H-2B visas to process shrimp, crab, crawfish, catfish, salmon, pollock and other finfish, in addition to working in commercial fishing and aquaculture in 2021 (*U.S. Department of Labor, in press*). A CDC analysis of data collected between March 1–May 31, 2020, found that Hispanic or Latino (72.8%), non-Hispanic Black (6.3%), and non-Hispanic Asian/Pacific Islander (4.1%) workers accounted for a disproportionately high number of COVID-19 cases in food manufacturing and agriculture, including seafood processing (*Waltenburg et al., 2021*). Foreign-born, migrant, and undocumented workers face additional challenges in the workplace, such as language barriers, fear of reprisal from employers, hesitancy to report illness or seek healthcare, crowded transportation and housing arrangements, and lesser access to food security, healthcare, and relief programs (*Bennett et al., 2020*; *Food and Agriculture Organization of the United Nations, 2020*; *National Guestwork Alliance, 2016*; *Waltenburg et al., 2021*). As noted above, initial rounds of federal funding and financial relief programs targeted participants in the commercial fishing and aquaculture sectors, while funds to enhance workplace safety and protect workers in the seafood processing sector have yet to be disbursed (*USDA, 2021*).

The World Bank estimates that women account for 42% of the workforce in fishing and post-harvest activities such as seafood processing in developed countries, though the proportion of women is likely higher if unpaid and informal labor were included

(*International Bank for Reconstruction & Development/The World Bank, 2012*; *UN Women, 2020*). It is suggested that women are differentially and disproportionately impacted by the COVID-19 pandemic, yet the extent to which the work, health, and wellbeing of women in the seafood industry have been affected by the pandemic is largely unknown (*Briceño-Lagos & Monfort, 2020*; *Wabnitz et al., 2021*). Women often fill low-wage, temporary, or casual labor positions that are more precarious and ineligible for government relief programs (*Bennett et al., 2020*; *Briceño-Lagos & Monfort, 2020*; *Wabnitz et al., 2021*), such as unemployment insurance and CARES Act funding. It has also been reported that mothers absorbed additional childcare responsibilities and reduced work hours because of closures of school and childcare facilities (*Alon et al., 2020*; *Collins et al., 2020*; *World Bank Group, 2020*), which in turn has the potential to affect fishing effort and landings. Furthermore, coping strategies adopted by women in fishing families include filling in as crew, producing value-added goods, engaging in shore-based employment, and direct marketing of seafood products (*Local Catch, 2021*; *Topness, 2021*). Systematic, disaggregated data collection coupled with intersectional research is needed to better assess the direct and indirect effects of the COVID-19 pandemic across diverse groups of workers within the seafood industry.

Most unanswered questions about how the pandemic has impacted fishers and the seafood industry pertain to the indirect effects on the industry. There is still limited guidance on how PPE should be worn on vessels specifically. The PPE that is worn on board is typically wet and can hold fish bacteria, allowing for exposure to occupational seafood respiratory allergy from biological and chemical agents that are associated with the work processes during processing, storing, preserving, and transporting seafood. There are physical factors other than seafood that pose a risk for respiratory illness symptoms as well, including hypertonic saline aerosols from wet environments. The exposure to these allergens comes from the inhalation of dust, steam, and seafood proteins generated from cleaning, scraping, cooking, or drying areas where seafood residues are located (*Jeebhay & Cartier, 2010*). It is not clear how these factors impacted breathing, respiratory illness contraction, and fishers developing more infections from the wet bacteria on their skin? There are also no reports on the number of accidents on board vessels during the timeline of the pandemic. Have accidents been happening more or less frequently within the seafood industry? Have seafood workers seen a difference in their stress levels, hours of sleep, and brain function because of the smaller crews? We hypothesize that mental and physical health as well as injury and mortality rates could all have suffered during the pandemic with smaller crew sizes and fewer workers on seafood processing lines, uncertainty about income, and changes in worker rules (*Sorensen, Echard & Weil, 2020*). However, smaller crew sizes and number of workers on seafood processing lines, as well as fishers leaving the industry, may have also led to decreases in injuries and fatalities. Several studies have now shown that some portions of the seafood sector have economically struggled because of the pandemic (*Bennett et al., 2020*; *Smith et al., 2020*; *Stoll et al., 2021*; *White et al., 2021*). Why did some fisheries rebound and quickly return to pre-COVID-19 demand and price levels, while others suffered significant loss and are

showing signs of slow recovery? How will these markets and the overall seafood pipeline look different after the pandemic? Moving forward, it is crucial to understand the effect of other COVID-19 variants or other future diseases on seafood workers in general.

There are several limitations to the current study. First, the data on COVID-19 cases across food systems was collected in an informal way. Specifically, the data was compiled by examining news articles and local health reports, where available. Ideally, this type of data would have been collected by a government agency to allow more consistent and detailed data collection. Because the data only consists of events which led to a significant outbreak, the data are also difficult to compare to numbers for the general population. Second, the news coverage analysis by key word returned over 20,000 news articles related to seafood workers and COVID-19. Future work could explore these articles more in depth as has been done in past work (*White et al., 2021*). And third, we were not able to directly examine how COVID-19 cases in the seafood industry varied by citizenship status, race, ethnic group, or gender. Additional work to examine potential discrepancies will be important for responding to future pandemics.

## CONCLUSIONS AND FUTURE WORK

The COVID-19 pandemic is another example of a significant shock for the fisheries and seafood industries. The seafood industry has a long history of shocks, including extreme weather events, fishery collapses, trade disruptions, and more (*Cottrell et al., 2019*; *Gephart et al., 2016*, *2017*; *White et al., 2021*). During any emerging event, relying on work on past shock events as well as ongoing data collection is critical. Because of a lack of timely information on the seafood sector, we had to rely on alternative data sources, including news articles and a user-contributed database of COVID-19 cases and outbreaks across the food sector. We hope this work serves as the foundation for other research on COVID-19 and seafood worker health and safety. Future work on COVID-19 and seafood could examine the disparate indirect effects of the pandemic and how they varied both in space and by sub-sector, how the seafood value chain has been changed overall, and how COVID-19 policies have affected seafood workers mental and physical health. Additional work could examine how differences in seafood distribution accelerated by the COVID-19 pandemic, including direct to consumer marketing (*Smith et al., 2020*; *Stoll et al., 2021*), may affect seafood workers moving forward. In addition, more general work around the role of shock events and the direct effects on all seafood workers is needed. This is important with increased demand for seafood products (*Costello et al., 2020*) occurring with an increased frequency and intensity of many rare events, such as hurricanes and heatwaves (*Bender et al., 2010*; *Cai et al., 2014*).

## ACKNOWLEDGEMENTS

We thank Leah Douglas at the Food and Environment Reporting Network for collating the data on COVID-19 cases within the U.S. food system. We also thank Julia Saltzman for designing figure 4.

### Funding

E.R.W. was partially funded by a grant from the Gund Institute for Environment at the University of Vermont. The funders had no role in study design, data collection and analysis, decision to publish, or preparation of the manuscript.

### Grant Disclosures

The following grant information was disclosed by the authors:
Gund Institute for Environment at the University of Vermont.

### Competing Interests

Julie Sorensen is employed by Bassett Healthcare Network.

### Author Contributions

- Easton R. White conceived and designed the experiments, performed the experiments, analyzed the data, prepared figures and/or tables, authored or reviewed drafts of the paper, and approved the final draft.
- Jill Levine performed the experiments, analyzed the data, prepared figures and/or tables, authored or reviewed drafts of the paper, and approved the final draft.
- Amanda Moeser analyzed the data, authored or reviewed drafts of the paper, and approved the final draft.
- Julie Sorensen analyzed the data, authored or reviewed drafts of the paper, and approved the final draft.

### Data Availability

   The data is owned by a third party which has given permission to this study's authors to use the data. Data are available upon request by contacting Sam Fromartz, sam@thefern. org, FERN's editor-in-chief.

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
