# Peer review of "The direct and indirect effects of a global pandemic on US fishers and seafood workers"

_PeerJ, doi:10.7717/peerj.13007_

## Round 0.1 · original submission · Major Revisions

Both reviewers thought the paper is a useful contribution, and could benefit from some additional clarity, mostly about the methods. In particular, Reviewer # 1 brought up several potential biases that should be addressed in the revised version.

Reviewer 1 ·

Basic reporting

no comment

Experimental design

The paper describes covid-19 case rates for the seafood production and processing sectors in the United States collected through media reports and secondary sources and compares these to other sectors of the food system. There are several areas where I have concerns, which could be addressed with more information about the methods and additional text for the readers about uncertainty.

I am concerned about reporting case rates to readers without adequately describing the potential error and uncertainty in both the numerator and the denominator. The numerator is based on i) cases reported by the media and ii) discovered by a search engine. Leah Douglas’s award-winning journalism is remarkable and I am glad you are able to put these data to use. I think the methods section needs a longer description of her methodology particularly since so much of the findings rest on this database. For the denominator, how close is the NAICS dataset to the actual number of workers? Have you vetted these values in other ways?

As for comparisons between the seafood and non-seafood sector, I would also like to hear more about how your team supplemented the FERN data with your own searches. Are you over-representing seafood? and would that make the seafood sector look riskier than non-seafood sectors when comparing head-to-head?

The characterization of the non-seafood sector as the rest of the food system is probably an overgeneralization. After looking at the appendix it appears to be just farms, food processors and meatpacking plants.

Validity of the findings

Introduction

“While occupational health has not been as widely explored as work-related injuries have in this sector…” I’m confused by this sentence. My understanding is that work-related injuries are part of occupational health?

“An article published in the Washington Post during the pandemic reported a sales decline as high as 95% for US fisheries (Reiley,2020)” A more robust source would be NOAAs recent covid-19 Report
https://www.fisheries.noaa.gov/feature-story/covid-19-impacts-us-fishing-and-seafood-industries-show-broad-declines-2020

Abstract
“Based on news reports, seafood workers were about twice as likely to contract COVID-19 as
workers in other parts of the overall US food system.” See comments below about concerns over this comparison.

Methods
“New articles” typo?

“This includes some articles published in the US about other countries as well as articles that might not focus on our specific topic.” I am a bit surprised that false positives were not filtered out? Can the author explain why?

Did you distinguish fishing vs aquaculture-related cases?

What is your definition of an outbreak?

Results

“Between March 2020 and September 2021, we identified a total of 1161 US news articles that specifically 156 mentioned (COVID) AND (worker OR employee) AND (seafood OR fisheries) (Fig. 1).” Was this from the initial 20,229 articles? In the methods it says these articles were not filtered, but it seems like they may have been. Please clarify.


“By May 2020, additional outbreaks had been reported across the country—from the Northeast to the Gulf Coast—primarily in seafood processing plants, but also at crawfish farms in Louisiana (Korban and Cherry, 2020).” The way this sentence is written, it sounds like covid cases were primarily within the seafood sector when we know they were in all aspects of society.


“Despite being a small part of the overall food system, the seafood industry accounted for 3.84% and 5.23% of all cases and outbreaks...” I am a little uncomfortable with this comparison because there may be bias. It appears you supplemented the FERN database with your own search strategy specifically for the seafood sector. Would that inflate the number of seafood cases relative to the denominator (total cases among farms, meatpackers and food processors)? In addition, was the study done of the food system or just the stages mentioned above in parentheses? It would be helpful to know what % of data points (rows) came from your search vs FERN.


“As a rough estimate, we estimated there were 65 COVID-19 cases 203 per 1000 workers in seafood versus 31 cases per 1000 for the overall food system.” Same concern as above. Were you oversampling the seafood sector? Is the food system being conflated with the production and processing sectors?

“These numbers are difficult to compare to the general population as the cases identified here in food systems include only publicly documented cases or outbreaks for workplaces, as opposed to cases in general” yes, I agree.

Discussion
“Seafood-related COVID-19 cases occurred throughout the seafood value chain, including fishing vessels, seafood processors, and seafood distributors.” Were the boundary conditions for the seafood sector search different than for FERN? I don’t think FERN tracked food distributors but I could be wrong.

I talked with seafood plant operators in Alaska and my understanding was that the summer processing season was challenging, to write and implement new covid protocols, but was fairly successful. It sounds like Copper River may be an exception?

Conclusions
I think this paragraph could be condensed by removing background literature and methods. A few sentences on the main findings and their importance is what I tend to look for.

Figure 2. what is the definition of a non-seafood case?

Table 1. how would the case rate compare to the meatpacking sector, which seems more similar than the general category of food processing?

The supplement only reports case data. Where is the outbreak data coming from? What is your definition of an outbreak?

Additional comments

no comments

·

Basic reporting

The language was clear and concise through most of the manuscript. In a few instances, the authors could help the readers by being more specific while making their points clearer. Additionally, increased specificity could help future researchers separate this study that focuses on COVID-19’s impact on the US seafood sector from the pandemic’s impact on the global seafood industry. Some of these instances are listed below:

Lines 56-59: In line 58 you describe the estimated “fatality rate for seafood workers was” which could read more clearly if you specified ‘seafood workers in Alaska was’. I recognize that you set the context and draw the conclusion in the first and last line, but this could be made clearer for a wider audience.
Lines 108-110: Despite potential repetitiveness, specifying that this manuscript focuses on the COVID-19’s impact on the American seafood sector could be accomplished by the insertion of “US” in the phrase “our goal was to document the spread of COVID-19 within the <US> seafood sector” (Line 109, suggested insertion indicated by text within <>)

Lines 114-115: “...seafood workers would be disproportionally affected by the pandemic compared to other seafood workers”. Did the authors mean ‘other food system workers’ or ‘other non-seafood workers’? Please correct this and it's recommended to double-check the manuscript for other instances where such a mix-up could impact the message of this manuscript.

Lines 126-129: The authors describe the search criteria, resultant number of articles, and caveats about some of the articles within this number. They then state that they “did not filter articles beyond this point.” For an international audience, it may be clearer to specify that articles published in the US but about other countries were filtered out, or use clearer language to describe the article filtering process.

Line 184: Given the context of this line at the start of a section on COVID-19 cases in the seafood industry, the wording “The first reported COVID-19 outbreak in the US occurred in April 2020 at a processing facility” could come across as slightly ambiguous to an international audience. Are the authors describing the first-ever reported COVID-19 outbreaks in the US? Was it at a seafood processing facility? Clarifications through more specific language could be helpful for a wider readership.

Lines 292-295: There seems to be a minor grammatical issue towards the end of this run-on sentence. This sentence could be modified to read as “... in an outbreak <where> the employers took no additional protection measures” (suggested word to be inserted in <>).

Lines 299-300: This sentence is somewhat unclear in terms of what the authors mean by ‘locally collected’ and ‘come from international countries’. The authors may want to clarify this statement.
Alongside the above instances requiring additional clarification and/or specification, the authors have done a commendable job in clearly describing additional details and caveats to their findings. Examples of these include lines 148-151 and 203-205.

The literature in this manuscript is current and well referenced. I particularly appreciated the literature described in the Introduction highlighting the dangerous working conditions of seafood workers, which were re-examined in the Discussion.

The tables and figures included in the manuscript are of professional quality. Code used to generate the figures and tables has been included and the data availability has been clearly described. A minor suggestion concerning Figure 2 is to change the faceting or plotting order of the plots so that the number of cases and outbreaks are represented as columns rather than rows. This would help readers track trends related to COVID-19 in seafood and non-seafood over time, as the different scales between the two systems don’t assist the comparison.

The article adheres to a standard manuscript structure. The inclusions of the case study comparisons and mention of additional hypotheses in the Discussion were effective.

The manuscript serves as a self-contained unit of publication where the goals and aims are clearly stated and concisely examined. The inclusion of additional questions for future studies to consider was beneficial to the manuscript's goals in highlighting considerations required for the wellbeing of workers in seafood systems.

Experimental design

This study attempted to document the spread of COVID-19 within the US seafood sector and also highlight the indirect effects of the pandemic. The authors examined news reports and covid case data in the US seafood industry to understand the geographic and temporal trends in cases, the indirect and direct effects of the pandemic, and how the US seafood sector compared to other US food sectors.

The knowledge gap concerning the health and working conditions of seafood workers during disease outbreaks and other diseases is described well in the Introduction, as are the replicable methods to address this gap. The use of secondary data and sensitive reporting of labor demographics and working conditions by the authors are at a high technical and ethical standard.

Validity of the findings

This manuscript’s reporting on the impacts of COVID-19 on the US seafood sector is timely and effective. The findings provided and additional questions to be asked are beneficial to fisheries socio-economic literature. Throughout the manuscript, the authors make the case for additional research attention on seafood workers and support these claims through background contexts, rigorously analyzed data, and logical conclusions. Caveats and additional details about the context and assumptions made by the author are made throughout the paper. In some instances, these could be more clearly specified (as has been discussed in detail under Basic Reporting). In summary, the conclusions are well stated, linked to the original research questions, and no new data or suppositions have been included.

Additional comments

No additional comments. Please refer to detailed comments in previous sections.

---

## Round 0.2 · accepted · Accept

Thank you for addressing the reviewer's comments -- the revised manuscript is significantly improved, and addresses the initial concerns about the analyses, etc.

Reviewer 1 ·

Basic reporting

the authors have addressed all of my concerns.

Experimental design

the authors have addressed all of my concerns.

Validity of the findings

the authors have addressed all of my concerns.

Additional comments

the authors have addressed all of my concerns.